# Chemical Composition of *Thymus leucotrichus* var. *creticus* Essential Oil and Its Protective Effects on Both Damage and Oxidative Stress in *Leptodictyum riparium* Hedw. Induced by Cadmium

**DOI:** 10.3390/plants11243529

**Published:** 2022-12-15

**Authors:** Viviana Maresca, Natale Badalamenti, Vincenzo Ilardi, Maurizio Bruno, Paola Bontempo, Adriana Basile

**Affiliations:** 1Department of Biology, University of Naples Federico II, 80126 Naples, Italy; 2Department of Biological, Chemical and Pharmaceutical Sciences and Technologies (STEBICEF), Università degli Studi di Palermo, Viale delle Scienze, Ed. 17, 90128 Palermo, Italy; 3Department of Precision Medicine, University of Campania “Luigi Vanvitelli”, Via L. De Crecchio 7, 80138 Naples, Italy

**Keywords:** *Thymus leucotrichus* var. *creticus*, essential oil, thymol, *p*-Cymene, antioxidant activity, DNA damage

## Abstract

The chemical profile of the essential oil (EO) of the aerial parts of *Thymus leucotrichus* var. *creticus* (Lamiaceae), a taxon not previously studied, was investigated by GC–MS analysis, using a DB–Wax polar column. Oxygenated monoterpenes and monoterpene hydrocarbons dominate the EO, with thymol (46.97%) and *p*-cymene (28.64%) as the main constituent of these two classes, respectively. The ability of the EO of *T. leucotrichus* to reduce Cd toxicity was studied in aquatic moss *Leptodictyum riparium*. To study EO-induced tolerance to Cd toxicity, apex growth, number of dead cells, DNA damage and antioxidant response in gametophytes were examined. The exogenous application of the EO yields a resumption of growth rate and a reduction in the number of dead cells; it also reduces the oxidative stress induced by Cd, as demonstrated by the reduction of the ROS content (with a decrease of 1.52% and 5%) and by the increased activity of antioxidant enzymes such as superoxide dismutase (SOD) (with an increase of 1.44% and 2.29%), CAT catalase (1.46% and 2.91%) and glutathione-*S*-transferase GST (1.57% and 1.90%). Furthermore, the application of the EO yields a reduction of DNA damage. These results clearly indicate the protective capacity of the EO of *T. leucotrichus* in modulating the redox state through the antioxidant pathway by reducing the oxidative stress induced by Cd.

## 1. Introduction

Cadmium is a well-known toxic element that damages the health of living organisms, therefore it represents an ecologically dangerous toxic metal. Given that Cd enters the food chain through plants, it is interesting to determine how plants respond to Cd. Plants growing in a growth medium with the addition of Cd show biochemical and physiological disorders such as growth inhibition, damage to membrane functions, alteration of ion homeostasis, decrease of water and nutrient transport, inhibition of photosynthesis, impaired metabolism, altered activities of several key enzymes and even cell death [1]. This results in excessive accumulation of reactive oxygen species (ROS) and methylglyoxal (MG), which can cause lipid peroxidation, protein oxidation, enzyme inactivation, DNA damage and interact with other plant cell constituents [2].

Cadmium can negatively affect plant growth, and its toxic effects might be apparent at both the morphological and physiological levels [3]. Nevertheless, the threshold of phytotoxic concentrations of Cd is very different across plants and depending on species, ecotypes, cultivars, etc. [4].

Many studies have shown that bryophytes are better than lichens and vascular plants at monitoring and tolerating heavy metal pollution in urban areas, as they are bioindicators and bio-accumulators of metals in the environment [5,6].

*Leptodictyum riparium* is an aquatic moss model used in environmental monitoring studies as it responds consistently to heavy metal-induced perturbations by activating a series of defense mechanisms. In particular, in recent years, the antioxidant response of moss to stress of both a pool of heavy metals and Cd alone has been studied [7,8].

The benefits that EOs have on health are already reported in ancient literature. Some of the purported beneficial functions of Eos—antiseptic, antioxidant and anti-inflammatory properties—have been supported by recent scientific investigation.

EOs have always been widely used for various purposes, not only as condiments for flavoring foods, ingredients in perfumes or in cosmetic applications, but also, and above all, for medical purposes, having demonstrated antibacterial, antifungal, virucidal, antiparasitic and insecticidal properties, as well as being a good analgesic, sedative and anti-inflammatory, hence being widely used in pharmaceutical industry.

When we speak of EOs, we are referring to volatile, natural compounds with a complex composition that are fat-soluble and soluble in organic solvents and which generally have a density lower than that of water. They are also characterized by a strong odor and are obtained from aromatic plants as secondary metabolites.

In nature, EOs play an important role in plant protection by virtue of their antibacterial, antiviral, antifungal, insecticidal and antioxidant properties.

The genus *Thymus* of the Lamiaceae family, contains more than 200 species distributed all over the world. It originates from the Mediterranean basin and is distributed also across Europe, Greenland, North America and Africa [9,10], and, due to its properties, *Thymus* ssp. have been largely employed in the food, cosmetics, perfume and pharmaceutical industries [11,12].

Due to their biological properties, the infusion and decoction of fresh or dried aerial parts of *Thymus* ssp. are used in ethnomedicine to treat numerous digestive and respiratory illnesses, such as colds, flu, indigestion, nausea and dysentery, and their use has been recently reviewed [13].

Non-volatile organic compounds detected in the extracts of *Thymus* ssp. include flavonoids, phenylpropanoids, lignans, tannins, organic acids, terpenoids and phytosterols. Several pharmacological studies showed that the extracts possess a large number of properties both in vitro and in vivo, including antimicrobial, antioxidant, antitumor, anti-inflammatory, analgesic, antispasmodic, antitussive, carminative, anti-hypertensive, anti-diabetic, anthelmintic activities, and so on [13].

By far, more investigations have reported on the EOs of *Thymus* ssp. that, in many cases, showed the large presence of two aromatic compounds, carvacrol and thymol, frequently accompanied by the couple *p*-cymene/*γ*-terpinene [14,15]. Other important components occurring in minor quantities are linalool, borneol and 1,8-cineole [16,17].

In addition, due to its antimicrobial and/or antioxidant compounds, the EOs of *Thymus* species have been utilized as alternatives to commercial synthetic chemicals in recent years. In fact, in order to extend the shelf-life of fresh foods, they have been incorporated into packaging materials [18,19,20], utilized as corrosion inhibitors for different metals in various acids [21] and applied in the disinfection of historical art and craft materials [22].

*Thymus leucotrichus* var. *creticus* (Bald.) Ronniger is a plant with frizzy woody primary branches bearing linear-lanceolate, sessile leaves that are gathered in axillary bundles and covered with hairs of variable length, with erect flower stems ascending up to 10 cm.

The inflorescence capitata range from ovoid to globose with bracts 1.5–3 mm wide, similar to leaves, and are purplish in color. Calyx 4.5–5.5 mm, with the upper teeth of 1–5 mm, are lanceolate and ciliated. Corolla is pinkish-purple, with the tube slightly exceeding the glass [23].

*Thymus leucotrichus* has a distribution that includes and goes beyond the Island of Crete, mainland Greece, Syria, Lebanon and Middle Eastern Turkey [24]. Within the species, two subspecies are distinguished: *T. leucotrichus* Hálacsy subsp. *leucotrichus* and *T. leucotrichus* subsp. *neiceffi* (Degen & Urum.) Jalas. Within *T. leucotrichus* subsp. *leucotrichus*, only *T. leucotrichus* var. *creticus* (Bald.) Ronninger is exclusive to Crete, and it is the subject of this work.

Consequently, in the frame of our ongoing research on endemic Mediterranean plants [25,26] and on the biological activity of EOs [27,28], we decided to investigate the EO composition of the aerial parts of *T. leucotrichus* var. *creticus*, a taxon not previously studied, as well as the antioxidant properties of its EO. This study focuses on the ability of the essential oil (EO) of *T. leucotrichus* to increase tolerance to Cd-induced oxidative stress in *L. riparium*. The purpose of this study is to evaluate the chemical composition of the EO of *T. leucotrichus* and its ability to induce a protective effect in *L. riparium* exposed to Cd stress: the growth rate, number of dead cells, levels of ROS, activity of antioxidant enzymes and DNA damage were evaluated.

## 2. Results and Discussion

### 2.1. Gas Chromatography and Mass Spectrometry (GC–MS) Analysis of the Essential Oil

The composition of the EO of *T. leucotrichus* var. *creticus* was analyzed by GC–MS analysis (as in Table 1). Fifteen compounds, divided into three classes, were identified and classified according to linear retention indices. In terms of compound classes, oxygenated monoterpenes (49.42%) dominate the EO, totally devoid of carvacrol and with thymol as the most abundant compound (46.97%). Monoterpene hydrocarbons are also dominant (45.51%), with *p*-cymene (28.64%) as main the constituent of the class. In contrast, sesquiterpene hydrocarbons accounted for only 3.07%, and no oxygenated sesquiterpenes were identified. Comparing the EO composition of *T. leucotrichus* from Turkey [29] to our results, we find it rich in thymol (37.01%), *p*-cymene (21.55%) and *γ*-terpinene (8.63%). On the other hand, the EO from *T. leucotrichus* plants collected in Bulgaria [30] showed a completely different profile; in fact, it was rich in sesquiterpene hydrocarbons (44.40%) and oxygenated sesquiterpenes (34.50%), with *β*-caryophyllene (23.10%), elemol (9.80%) and germacrene D (6.50%) as the main constituents, and quite poor in thymol (2.7%). In addition, the two accessions of *T. leucotrichus*, collected in Greece (Mt. Parnon, Pelloponesus, and Mt. Dirfi, Evoia) [31], proved to be very rich in sesquiterpenes with *β*-caryophyllene (13.2% and 17.5%, respectively) as the principal metabolite. The co-occurrence of thymol and *p*-cymene, as principal metabolites, was also observed in some other *Thymus* taxa such as *T. vulgaris* L. from Egypt [32], *T. glandulosus* Lag. from Morocco [33], *T. pulegioides* L. from Southern Italy [34], *T. munbyanus* Boiss et Reuter [syn. *T. ciliatus* (Desf.) Benth.] [35,36], *T. guyonii* De Noe from Algeria [35], *T. transcaucasicus* Ronniger [37], *T. trauveterri Klokov* & Des.-Shost. [38], *T. daenensis* Čelak [39] and *T. migricus* Klokov & Des.-Shost. [40] from Iran.

### 2.2. Inhibition of the Growth Rate

The EC_50_ was tested on *L. riparium* gametophytes exposed to Mohr’s solution containing CdCl_2_ concentrations ranging from 0.5 to 20 mM with a progressive increase of 0.5 M for 7 days in both EO-pretreated and non-pretreated samples. From toxicity tests, *L. riparium* was found to be a very resistant species, with estimated EC_50_ values (for a 7-day test period) of 4.5 mM in the samples without pretreatment; by contrast, the samples that were pretreated with the EO of *T. leucotrichus* showed a significantly higher EC_50_, reaching a concentration of 11.5 mM for the samples pretreated with 0.16% EO and 18 mM for samples treated with 0.4% EO (Appendix A). It is evident that the pretreatment with EO, which provides protection from the damage exerted by the metal, requires greater use of Cd to obtain the considered toxic effect. The effect of the different concentrations tested served to choose the optimal concentration to test the protective effect of the EO against cadmium stress.

### 2.3. Percentage of Dead Cells

The samples exposed to 1.5 mM of CdCl_2_ without treatment with EOs showed a number of damaged cells after 7 days of culture, with more or less evident plasmolysis of 12 ± 0.3%; while the samples pretreated with EO reached, respectively, only 2.30 ± 0.80% for the pretreated samples with the concentration of 0.16% EO and 1.10 ± 0.20% for the samples pretreated with 0.4% EO (Table 2). The protective effect of the EO on the survival of cells treated with CdCl_2_ was therefore evident.

### 2.4. Detection of ROS and Antioxidant Activity Enzyme

As can be seen from Figure 1, after exposure of *L. riparium* to 1.5 mM of CdCl_2_, an increase in ROS is observed in the samples without pretreatment, while a decrease is observed in samples pretreated with *T. leucotrichus* EO. In particular, *L. riparium* samples pretreated with 0.4% of EO show a drastic reduction of ROS both compared to samples pretreated with CdCl_2_ and to samples without pretreatment. However, a statistically significant reduction is also observed in samples pretreated with 0.16% EO compared to samples without pretreatment.

Regarding the antioxidant activity, evaluated through the activity of the SOD, CAT and GST enzymes, a significant increase of all three enzymes is observed in the samples pretreated with the *T. leucotrichus* EO compared to the samples without pretreatment; in particular, the 0.4% concentration of the EO seems to show a greater effect. This increase in enzyme activity probably explains why a decrease in ROS is observed in EO pretreated samples. Activation of antioxidant enzymes is an intrinsic defense strategy to adjust the ROS contents of cells according to the metabolic needs at a specific time.

### 2.5. Comet Assay

Figure 2 shows DNA damage following exposure to Cd in terms of DNA damage, tail moment and olive moment in both pretreated and untreated samples. Samples of *L. riparium* exposed to 1.5 mM of CdCl_2_ show an increase in all three parameters taken into consideration. This should not be surprising given that an excess of ROS can, among other effects, also cause DNA damage, including its breakdown, which, however, can also be due to a direct effect of heavy metals on the nucleotide [41].

Samples pretreated with *T. leucotrichus* EOs showed less damage than non-pretreated samples, possibly due to a protective action of the EOs.

Interestingly, even in the case of protection against DNA damage, the 0.4% EO concentration was found to have a greater protective action.

It is known that essential oils are used for healing purposes based on their many properties. On the other hand, there are few data regarding a protective effect of essential oils on stress from pollutants and, specifically, from heavy metals.

With these results it is possible to hypothesize that *T. leucotrichus* EO can somehow counteract the oxidative stress induced by CdCl_2_ and consequently limit DNA damage. However, the evidence from these studies needs to be confirmed by further experiments.

## 3. Materials and Methods

### 3.1. Essential Oil

The aerial parts of *T. leucotrichus* var. *creticus* were collected along the road from Kolimpari to Afrata, North Crete, Greece (35°34′08.66″ N, 23°46′24.73″ E, 150 m m.s.l.), in June 2022. A voucher specimen has been deposited in the STEBICEF Department, University of Palermo (PAL113454).

A total of 100 g of the aerial parts of *T. leucotrichus* var. *creticus* were subjected to hydrodistillation for 3 h second using Clevenger’s apparatus [42]. The oil, a yield 2.48% (*v*/*w*), was dried with anhydrous sodium sulphate, filtered and stored in the freezer at −20 °C, until the time of the analyses.

### 3.2. GC–MS Analysis of Essential Oil

Analyses of essential oils were performed according to the procedure reported by Rigano et al. [43].

### 3.3. Plant Material and Heavy Metal Treatment

Field-grown moss *L. riparium* Hedw (Amblystegiaceae) was collected in the Botanical Garden of the University of Naples Federico II, Italy. Approximately 1 g of the samples was rinsed with sterile distilled water and inoculated into flasks containing sterile modified Mohr’s medium [44] and cultured for 7 days (acclimatization). After that, two concentrations of EO in ethanol solutions, at 0.16% and 0.4% (*v*/*v*), were applied as foliar spray on the gametophytes for 7 days. Subsequently, the plants that were pretreated with and without the EO were irrigated with Mohr solution containing 1.5 mM CdCl_2_ for 7 days in a climate-controlled room with a temperature ranging from 13 to 20 °C (night/day), 70% relative humidity and a photoperiod of 16 h light (40 μEm^−2^ s^−1^)/8 h dark.

### 3.4. Inhibition of Growth Rate

We determined ErC_50_ (the concentration at which a 50% inhibition of growth rate is observed) as the endpoint for ecotoxicity. Total frond count (carried out on 1 g of moss) was used to monitor growth at metal concentrations between 0.5 and 20 mM, which was the range in which the plants remained viable and were able to regenerate damaged tissues. The total frond count was defined as the number of new formed shoots. Growth was monitored every day for 7-day test period by counting fronds under a magnifying glass. From these values, growth was determined as described in Basile et al. [45].

The effect of the different concentrations tested served to choose the optimal concentration to test the protective effect of the EO against Cadmium stress. The concentration of 1.5 mM was chosen as it is effective in determining a toxic and responsive effect but is far from the EC_50_ (which we consider excessively toxic). finally, this choice is also justified by the fact that it shows concentrations close to it in cadmium-polluted watercourses, therefore a realistic situation, in which moss may find itself having to survive [46].

### 3.5. Percentage of Dead Cells

The percentage of dead cells was calculated by light microscope observations made on moss gametophytes with toluidine blue stained semi thin sections, prepared as reported in Basile et al. [6], on samples treated without EO, 0.16% and 0.4% EO, and after with CdCl_2_.

### 3.6. Detection of ROS and Antioxidant Activity Enzyme

A total of 0.5 g of moss was homogenized with 0.1 mL of 50 mM potassium phosphate-buffered solution (PBS) (pH 7.4) using a sterile pestle. The protein extract was used to evaluate the levels of ROS and the activity of the antioxidant enzymes CAT, SOD and GST [8].

### 3.7. Comet Assay

The moss (0.5 g) was gently sliced using a fresh razor blade. The plate was kept tilted on ice so that the isolated nuclei would collect in a cold Tris buffer. The protocol was performed as reported by Maresca et al. [8].

### 3.8. Statistical Analysis

ROS production and SOD, CAT and GST enzyme activities were examined by one-way analysis of variance, followed by Tukey’s multiple comparison post-hoc test. In all figures, values are presented as mean ± st. err; numbers not accompanied by the same letter are significantly different at *p* < 0.05. Data were analyzed using the software Statistical, version 7.0 (StatSoft, Tulsa, OK, USA).

## 4. Conclusions

The present study has focused on determining the yield, chemical composition and ability of EO of *T. leucotrichus* to increase tolerance to Cd-induced oxidative stress in *L. riparium*.

Among natural plant products, EOs deserve special attention due to their use. EOs, in fact, are used for multiple purposes, such as personal and home care, often in food, as human and animal repellents and for the treatment of various diseases. Despite the differences in the chemical composition of EOs obtained from different plants with different extraction methods, their major constituents belong to the same chemical classes, such as mono- and sesquiterpenes, aldehydes, ketones, ethers and esters, alcohols and hydrocarbons. The presence of these compounds yields both chemical–physical and biological properties such as antibacterial, antifungal, antioxidant, anti-inflammatory and antitumor activity in numerous cellular and animal models. Furthermore, currently, the distillation of EO from different plant organs is a reliable and economical process. As far as their efficacy is concerned, numerous studies have documented the biological activity of EOs as well as clarifying their mechanism of action and pharmacological targets. However, the paucity of studies on the protective capacity against heavy metals on possible plant targets limits the potential of EOs as effective and safe phytoprotective agents. More specific and in-depth studies are, therefore, needed to achieve a high level of scientific evidence and ascertain the real efficacy and safety of plant products.

## Figures and Tables

**Figure 1 plants-11-03529-f001:**
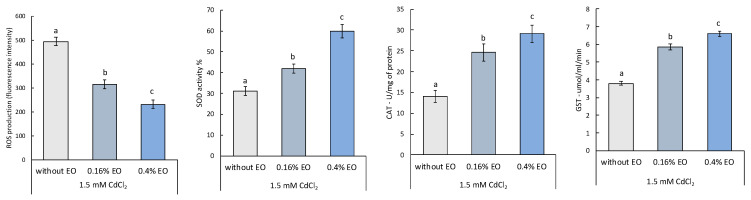
ROS amount and antioxidant/detoxifying enzyme activities (SOD, CAT and GST) in *L. riparium* gametophytes treated without EOs, with 0.16% and 0.4% of Eos, and after with CdCl_2_. Bars not accompanied by the same letter were significantly different at *p* < 0.05. Data are mean of three independent experiments ± SE (n = 5).

**Figure 2 plants-11-03529-f002:**
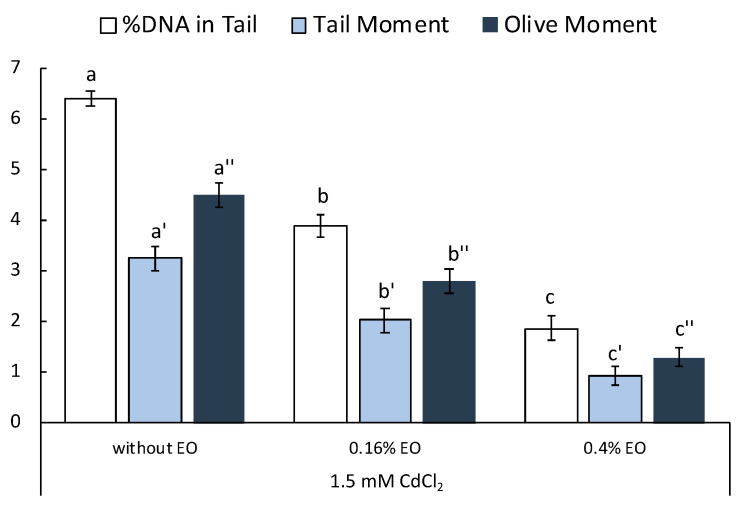
Comet assay results (DNA damage, tail moment and olive moment) in *L. riparium* gametophytes treated without EOs, 0.16% and 0.4% of EOs, and after with CdCl_2_. Data were presented as mean and standard error and they were analyzed with a paired *t*-test. Bars not accompanied by the same letter were significantly different at *p* < 0.05.

**Table 1 plants-11-03529-t001:** Chemical composition of *Thymus leucotrichus* var. *creticus* essential oil collected in Greece.

No.	Compounds ^a^	LRI ^b^	LRI ^c^	Area (%)	Ident. ^d^
**1**	*α*-Pinene	1009	1017	1.95	1, 2
**2**	Camphene	1053	1060	2.90	1, 2
**3**	*β*-Pinene	1095	1099	1.08	1, 2
**4**	Sabinene	1109	1111	0.14	1, 2, 3
**5**	3-Carene	1112	1114	0.06	1, 2, 3
**6**	*α*-Phellandrene	1163	1177	0.20	1, 2, 3
**7**	4-Carene	1155	1157	3.14	1, 2
**8**	Limonene	1156	1060	0.39	1, 2, 3
**9**	*γ*-Terpinene	1241	1248	7.01	1, 2
**10**	*p*-Cymene	1267	1278	28.64	1, 2, 3
**11**	*β*-Linalool	1548	1557	0.89	1, 2
**12**	*β*-Caryophyllene	1595	1608	2.83	1, 2, 3
**13**	Isoborneol	1655	1660	1.56	1, 2
**14**	*α*-Caryophyllene	1679	1687	0.24	1, 2, 3
**15**	Thymol	2123	2139	46.97	1, 2, 3
	**Monoterpene Hydrocarbons**			**45.51**	
	**Oxygenated Monoterpenes**			**49.42**	
	**Sesquiterpene Hydrocarbons**			**3.07**	
	**Total**			**98.00**	

^a^ Components listed in order of elution on an DB–Wax column; ^b^ Linear retention index on a DB–Wax polar column; ^c^ Linear retention indices based on the literature (https://webbook.nist.gov/, accessed on 6 November 2022). ^d^: 1 = retention index identical to bibliography; 2 = identification based on comparison of MS; 3 = retention time identical to authentic compounds.

**Table 2 plants-11-03529-t002:** Percentage of dead cells from the total within the *L. riparium* gametophytes treated without EO, with 0.16% and with 0.4% of EO, and, after, with CdCl_2_.

	CdCl_2_ 1.5 mM		
	without EO	CdCl_2_ + 0.16% EO	CdCl_2_ + 0.4% EO
**Percentage of death cells**	12% ± 0.3	2.3% ± 0.8	1.1% ± 0.2

## Data Availability

Not applicable.

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
