# Peer review of "Chemical Composition of Thymus leucotrichus var. creticus Essential Oil and Its Protective Effects on Both Damage and Oxidative Stress in Leptodictyum riparium Hedw. Induced by Cadmium"

_plants, 2022, doi:10.3390/plants11243529_

Round 1

Reviewer 1 Report

I think that the authors have created a rather large and significant manuscript. The manuscript covers an interesting topic. The lack of studies on the protective capacity against heavy metals on possible plant targets limits the potential of EOs as effective and safe phyto-protective agents. I suggest that the manuscript may be accepted for publication as an original scientific paper. 

Author Response

We thank Reviewer 1 for the rewarding comment. We are honored and delighted that Reviewer 1 enjoyed our manuscript

Reviewer 2 Report

All the comments are exist in the attached file

Author Response

We thank Reviewer 2 for the careful and fair observations. We have modified the text according to your indications. The manuscript is certainly clearer and thanks to your suggestions. The changes are indicated in yellow in the text.

Reviewer 3 Report

First of all, I would like to suggest the authors to be less self-citing.. 17 out of 51 references are papers from some of the authors. Self-citing is not a good habit.

M&M: the paragraph 2.2. Inhibition of the growth rate should clearly state which kind of stress is considered… mM of what? CdSO4? CdCl2? The reader understands that the authors used CdCl2 just in paragraph 2.3. Evaluating EC50 for heavy metals should always be contextualized with the conditions of the treatments.. Plants respond differently to Cd depending on the concentrations of other nutrients in the growth media. Authors should give more details here, not only in M&M. (be careful that in Table 2 1.5M Cd is indicated, not 1.5mM)

It is not clear how much material has been used for each experiments.

I would also like to see the effect of EO alone, without Cd treatment, on the moss, just to see if there are effects on cells of the EO treatment.

Then it would also be interesting to see pictures of the moss, and the phenotype upon Cd treatments.

Round 2

Reviewer 2 Report

Dear editor thank you for this review this manuscript is good for readers and can be accepted after minor revision

Reviewer 3 Report

I've no comments, more than those I've already raised.